# The Relationship between Dietary Habits and Frailty in Rural Japanese Community-Dwelling Older Adults: Cross-Sectional Observation Study Using a Brief Self-Administered Dietary History Questionnaire

**DOI:** 10.3390/nu10121982

**Published:** 2018-12-14

**Authors:** Kayoko Tamaki, Hiroshi Kusunoki, Shotaro Tsuji, Yosuke Wada, Koutatsu Nagai, Masako Itoh, Kyoko Sano, Manabu Amano, Hatsuo Maeda, Yoko Hasegawa, Hiromitsu Kishimoto, Sohji Shimomura, Ken Shinmura

**Affiliations:** 1Division of General Medicine, Department of Internal Medicine, Hyogo College of Medicine, Nishinomiya, Hyogo 663-8501, Japan; ka-tamaki@hyo-med.ac.jp (K.T.); kusunoki1019@yahoo.co.jp (H.K.); 2Department of Orthopedic Surgery, Hyogo College of Medicine Sasayama Medical Center, Sasayama, Hyogo 669-2321, Japan; tj13041305sho@gmail.com; 3Department of General Medicine and Community Health Science, Hyogo College of Medicine Sasayama Medical Center, Sasayama, Hyogo 669-2321, Japan; yu-wada@hyo-med.ac.jp (Y.W.); s-shimo@hyo-med.ac.jp (S.S.); 4Department of Rehabilitation, Hyogo College of Medicine Sasayama Medical Center, Sasayama, Hyogo 669-2321, Japan; 5School of Rehabilitation, Hyogo University of Health Sciences, Kobe, Hyogo 650-8530, Japan; nagai-k@huhs.ac.jp (K.N.); masakoitoh@huhs.ac.jp (M.I.); kyokosano@huhs.ac.jp (K.S.); 6School of Pharmacy, Hyogo University of Health Sciences, Kobe, Hyogo 650-8530, Japan; mbam@huhs.ac.jp (M.A.); hmaeda@huhs.ac.jp (H.M.); 7Department of Dentistry and Oral Surgery, Hyogo College of Medicine, Nishinomiya, Hyogo 663-8501, Japan; cem17150@dent.niigata-u.ac.jp (Y.H.); kisihiro@hyo-med.ac.jp (H.K.)

**Keywords:** dietary fiber, dietary habit, frailty, mineral, older adults, sex difference, vitamin

## Abstract

To develop effective nutritional interventions for preventing frailty, the specific problems associated with the dietary habits of individuals based on sex differences must be identified. The purpose of this study was to evaluate the association between dietary habits and frailty in rural Japanese community-dwelling older adults. We recruited 800 participants, aged 65 and older, who underwent a comprehensive health examination between November 2015 and December 2017. Dietary habits were assessed by a brief self-administered dietary history questionnaire. Frailty was determined using either the Kihon Checklist (KCL) or the Japanese version of the Cardiovascular Health Study (J-CHS). The percentage of frail older adults was 8.4% according to KCL and 4.0% according to J-CHS. Various kinds of nutrient intakes, including three major nutrients, minerals, and vitamins in frail men, according to KCL, were the lowest. By contrast, there were no differences in nutrient intake between the robust, prefrail, and frail female groups according to KCL. We found significant associations of the intakes of soluble dietary fiber, potassium, folate, and vitamin C with a frail status in men (*p* = 0.035, 0.023. 0.012, and 0.007, respectively), and an association of the intake of vitamin C with a frail status in women (*p* = 0.027) according to J-CHS. Attention should be paid to the diagnostic criteria of frailty and to sex differences, when nutritional interventions for the prevention of frailty are planned.

## 1. Introduction

In aged societies, such as Japan, the establishment of management protocols for age-related diseases and long-term care, in connection with disability, are urgently required. In recent years, providing appropriate interventions for the elderly at the time of frailty onset has been considered important in the prevention of the progression of disability, as is the need for long-term care [1,2]. In the frailty cycle, the onset of muscle mass reduction (that is, presarcopenia) associated with aging and comorbidities, leads to muscle weakness and/or a decline in physical function (sarcopenia) as well as physical activity, which may result in the reduction of total energy expenditure [1,2,3]. The decrease in energy requirements also leads to the reduction of appetite and incidences of malnutrition and, eventually, reduction in the muscle mass, resulting in a vicious cycle of frailty [1,2,3]. Therefore, appropriate interventions should be implemented according to the phase of the frailty cycle.

To prevent the progression of frailty, interruption and slowing down of this vicious cycle are necessary. Lower nutritional status has been responsible for the progression of frailty [4,5]. Thus, appropriate interventions, in terms of diet and dietary behaviors, are necessary before the condition progresses to an irreversible status that requires long-term care. Previous studies have reported that increased protein intake and sufficient calorie intake are useful for the prevention and improvement of frail conditions [6,7,8,9,10,11]. In addition, high protein intake enhanced the protein anabolic response in frail individuals, resulting in an improvement in the negative nitrogen equilibrium [12]. To prevent age-associated reduction in lean mass, the recommended dietary allowance of protein of 0.8–1.0 g/kg/day is insufficient—a protein allowance of 1.0–1.3 g/kg/day is advisable [13]. Furthermore, a recent meta-analysis of observational studies demonstrated that low protein intake is closely associated with frailty in older adults [9,14]. In addition to proteins, micronutrients—such as vitamin D, which is associated with bone metabolism, vitamins B6, C, and E, and folate that exerts antioxidant effects—were involved in frailty [6,9,15]. However, evidence regarding the association between dietary habits and frailty, which will help to establish appropriate nutritional interventions against frailty, is still lacking.

Frailty may be targeted to extend the healthy lifespan in community-dwelling older adults living at home. We hypothesized that nutritional interventions against frailty should be established, based on sex differences in dietary habits. Therefore, this cross-sectional observation study aimed to clarify the association between dietary habits and frailty diagnosed by two different methods in rural Japanese community-dwelling older adults using a self-administered dietary history questionnaire.

## 2. Materials and Methods

The cross-sectional study was designed as a Frail Elderly in the Sasayama-Tamba Area (FESTA) study. The study population, comprising of individuals ≥65 years old, was recruited among community-dwelling elderly individuals in the Sasayama-Tamba area—a rural area in the Hyogo prefecture, Japan—between November 2015 and December 2017. This study was conducted in accordance with the Declaration of Helsinki, and the institutional review board of Hyogo College of Medicine approved the FESTA study, which complies with the current laws of Japan (approval No. Rinhi 0342). The survey commenced after an oral/written explanation of the study objectives, methods, and expected outcomes was provided to participants. All participants provided written informed consent. The data used in this study were anonymized and masked for analysis.

### 2.1. Exclusion Criteria

Of the 844 participants, we excluded individuals who displayed decreased cognitive function (Mini-Mental State Examination (MMSE) score less than 24) (males *n* = 7, females *n* = 16). In addition, individuals whose total energy intake, evaluated by a brief self-administered dietary history questionnaire (BDHQ), was less than 600 kcal/day or more than 4,000 kcal/day were also excluded from further analyses (males *n* = 11, females *n* = 10). Therefore, 800 individuals were included in our analysis. The Kihon Checklist (KCL) analysis was performed in 800 participants (males *n* = 254, females *n* = 546). Analysis for the Cardiovascular Health Study (CHS) was performed in 796 older adults (males *n* = 252, females *n* = 544) after excluding individuals who failed to complete physical performance and movement tests (males *n* = 2, females *n* = 2). Analysis for the Asian working group for sarcopenia (AWGS) was performed in 788 older adults (males *n* = 250, females *n* = 538) after excluding individuals with electric devices implanted, including a pacemaker and a cardioverter-defibrillator (males *n* = 2, females *n* = 6).

### 2.2. Measurement of Body Composition

To evaluate muscle mass, participants were subjected to a bio-electrical impedance analysis (BIA) using InBody770 (Biospace Japan). Skeletal mass index (SMI) was defined as the muscle mass of extremities divided by the height squared. Testing was performed in the morning.

### 2.3. Diagnosis of Frailty

For the evaluation of frailty, we used two distinct criteria. A self-administered questionnaire survey was conducted using the KCL, which consists of 25 questions, and is used in screening participants who require care prevention in Japan [16]. KCL is divided into the following domains: instrumental activities of daily living (ADL), social ADL, exercise, falling, nutrition, oral function, cognitive function, and depression. Participants were asked to respond either “negative” (score: 1) or “positive” (score: 0), for a total score of 25. Frailty was determined using the methods introduced by Satake et al. [16]: frail, 8–25 points; pre-frail, 4–7 points, and robust, 0–3 points.

We also characterized the frailty phenotype according to limitations in three or more of the following five conditions based on CHS: slow gait speed, weakness, exhaustion, low activity, and weight loss [17]. We scored the number of corresponding components as the frailty score, after modification (Japanese version of the CHS: J-CHS) [18]. Participants who did not have any of these components were considered as non-frail (robust), and those with one or two components were considered as prefrail. Slow gait speed was established based on a cutoff of <1.0 m/s. Weakness was defined using maximum grip strength and was established according to a sex-specific cutoff (<26 kg for men and <18 kg for women). Exhaustion was considered present if a participant responded “yes” to the following question included in the KCL: “In the last 2 weeks, have you felt tired without a reason?” We evaluated physical activity by asking the following questions about the time spent engaged in exercise: “Do you engage in low levels of physical exercise aimed at health?” If participants answered “no” to the questions, we classified them to the low activity category. Weight loss was assessed by a response of “yes” to the question, “Have you lost 2 kg or more in the past 6 months?”

### 2.4. Dietary Survey

Dietary habits during the preceding month were assessed using a brief-type self-administered diet history questionnaire (BDHQ)] [8,19]. The BDHQ for older people is a 10-page structured questionnaire used to inquire about the consumption frequencies of selected foods commonly consumed in Japan, general dietary behavior, and usual cooking methods. The daily intakes of 58 food items, energy, and selected nutrients were calculated using the BDHQ responses and an ad hoc computer algorithm based on the Standard Tables of Food Composition in Japan [20]. For all study participants, BDHQ calculations were made using fixed, sex-specific portion sizes. The methods used to calculate the dietary intakes and the validity of the BDHQ have been detailed elsewhere [8,19]. Energy intake distribution was determined based on the calculated nutrient intake by using the residual method. All participants answered a BDHQ by themselves, and a trained managerial dietician or investigator helped them complete all questionnaires.

### 2.5. Statistical Analysis

The results were expressed as mean ± standard deviation for continuous variables. Categorical variables were expressed as percentages. To compare participants’ characteristics, an unpaired *t* test and a *χ*^2^ test were used for comparisons of continuous and categorical variables, respectively. For intergroup comparisons, data were analyzed using a *χ*^2^ test and one-way analysis of variance (ANOVA), followed by the Tukey–Kramer test. To determine significant differences in 4 dietary nutrients (soluble dietary fiber, potassium, folate, and vitamin C) between robust, prefrail, and frail individuals after excluding the possible confounding effects of baseline differences, an analysis of covariance (ANCOVA), adjusting for age, BMI, smoking habit, and alcohol consumption, was conducted. For each statistically significant effect of a dietary nutrient, the Bonferroni post hoc test was executed. *p*-values < 0.05 were considered statistically significant. JMP 13.1 software was used for data analysis.

## 3. Results

### 3.1. Basic Characteristics and the Prevalence of Frailty

The average age of the 800 participants was 72.6 ± 5.8 years. Of these, 254 were male (31.8%) and 546 were female (68.3%); 529 (66.1%) were young-old (65–74 years old) and 271 (33.9%) were old-old (≥ 75 years old). As shown in Table 1, there was no significant difference between men and women in terms of age, blood pressure, MMSE score, and exercise habits. However, height, body weight (BW), and body mass index (BMI), were higher in men than in women. There was a higher number of current smokers and alcohol drinkers among men. We also found differences between sexes in the proportions of comorbidities, such as hypertension, diabetes mellitus (DM), dyslipidemia, cardiovascular disease (CVD), osteoporosis, and a history of cancer.

As for frailty severity, in this study, 59.5% of the participants were healthy (robust), 32.1% prefrail, and 8.4% frail, according to the KCL criteria, and 41.6% were robust, 54.8% prefrail, and 4.0% frail, according to the J-CHS criteria. The concordance rate for the two diagnoses was 57.0%. In terms of the diagnosis of sarcopenia, 68.8% of the participants were robust, 27.5% had presarcopenia, and 3.7% had sarcopenia. Although there was no difference in the prevalence of frailty between men and women, the prevalence of sarcopenia was significantly higher in women than in men (Table 1).

There were significant differences between men and women in terms of body size, as well as smoking and alcohol consumption status. Therefore, nutritional analyses were performed separately in men and women. In addition, we compared basic characteristics, nutrient intake, and dietary habits between robust, prefrail, and frail individuals, because the prevalence of sarcopenia in men was too low to obtain meaningful analysis results.

### 3.2. Comparison of the Basic Characteristics, Nutrient Intake, and Dietary Habits between Robust, Prefrail, and Frail Individuals, Diagnosed by the KCL Criteria

As shown in Table 2, the prefrail male group, diagnosed by the KCL criteria, was older, compared with the robust male group. However, there were no differences in other basic characteristics between the three groups.

The carbohydrate, total protein (both animal and vegetable) and fat (both animal and vegetable) intake in the frail male group was the lowest among the three groups (Figure 1A–E). In addition, the intake of dietary fiber, both soluble and insoluble, was the lowest in the frail male group (Figure 1F). Among various nutrients, the intakes of sodium, potassium, calcium, magnesium, phosphorus, iron, zinc, α-tocopherol, vitamins B1 and B2, and pantothenic acid were the lowest in the frail male group (Table 2). The intake of copper, β-carotene (equivalent), γ-tocopherol, vitamins B6, C, and K, folate, and cryptoxanthin were lower in the frail male group than in the robust male group. In terms of food groups, the intake of sugar and sweeteners, and dairy products was significantly lower in the frail male group than in the prefrail male group.

As shown in Table 3, age was the highest and height was the lowest in the frail female group. Age was significantly higher in the prefrail female group than in the robust female group. In addition, the frail female group showed lower alcohol consumption and lower exercise status. The proportion of individuals with chronic kidney disease (CKD), gastrointestinal disease, asthma/chronic obstructive pulmonary disease (COPD), osteoporosis, and a history of cancer was significantly higher in the frail female group, compared with the robust female group.

The plant fat intake was significantly higher in the prefrail female group than in the robust female group (Figure 1E). There was no significant difference in the intake of other nutrients between the three groups (Figure 1A–F and Table 3). In terms of food groups, there was no significant difference among the 3 groups, except that oil intake was higher in the frail female group than in the robust female group.

### 3.3. Comparison of the Basic Characteristics, Nutrient Intake, and Dietary Habits between Robust, Prefrail, and Frail Individuals, Diagnosed by the J-CHS Criteria

As shown in Table 4, no statistically significant differences in the basic characteristics were observed between the robust, prefrail, and frail male groups, diagnosed by the J-CHS criteria, except for the exercise status.

The carbohydrate and plant protein intake tended to be lower in the frail male group than in the robust male group, but the difference did not reach statistical significance (*p* = 0.064, 0.092, respectively) (Figure 2A,C). Although the difference in total dietary fiber intake between the robust and frail groups was also marginal (*p* = 0.059), the intake of soluble dietary fiber was significantly lower in the prefrail male group than in the robust male group (Figure 2F). Among various nutrients, the intake of potassium, iron, β-carotene (equivalent), vitamins B1, B2, B6, and C, folate, and pantothenic acid was lower in the prefrail male group than in the robust male group (Table 4). The intake of vitamin B2 and B6 was lower in the frail male group than in the robust group. In terms of food groups, the intake of confectioneries was significantly lower in the frail male group than in the robust male group.

As shown in Table 5, age was the highest and the height was the lowest in the frail female group between the three groups. BMI was significantly higher in the prefrail female group, compared with that in the robust female group. In addition, the frail female group had the lowest exercise status. The proportion of chronic liver disease, CKD, and gastrointestinal disease was significantly higher and that of dyslipidemia was significantly lower in the frail female group.

No statistically significant differences in carbohydrate, protein, and fat intake were observed between the three groups (Figure 2A–E). Although the difference in total dietary fiber and insoluble dietary fiber intake between the robust and prefrail female groups was marginal (*p* = 0.080, 0.084, respectively), the intake of soluble dietary fiber was significantly lower in the prefrail female group, compared with the robust female group (Figure 2F). Among various nutrients, the intake of potassium, vitamin C, and folate was lower in the prefrail female group than in the robust female group (Table 4). In terms of food groups, the intake of eggs was the highest in the frail female group. By contrast, the intake of meats tended to be lower in the frail female group (*p* = 0.098).

### 3.4. Reevaluation of Significant Associations of the Intakes of Soluble Dietary Fiber, Potassium, Folate, and Vitamin C with a Frail Status by ANCOVA

Since decreased intakes of soluble dietary fiber, potassium, folate, and vitamin C were commonly observed in frail men by the KCL criteria, in prefrail men by the J-CHS criteria, and in prefrail women by the J-CHS criteria, we conducted an ANCOVA to determine significant differences in 4 dietary nutrients between the robust, prefrail, and frail groups, after excluding the possible confounding effects of baseline differences. An analysis of men revealed significant associations of the intakes of soluble dietary fiber, potassium, folate, and vitamin C with a frail status diagnosed by the J-CHS (*p* = 0.035, 0.023. 0.012, and 0.007, respectively) and that diagnosed by the KCL (*p* = 0.008, 0.011. 0.007, and 0.006, respectively). In contrast, we did not observe associations of a frail status with any dietary factors except vitamin C in women (*p* = 0.128, 0.137. 0.107, and 0.027, respectively). Thus, we concluded that the sufficient intakes of dietary fiber, potassium, folate, and vitamin C may be critical for preventing frailty in men.

## 4. Discussion

This study provides several novel and interesting findings, with regards to the dietary habits of older adults with frailty, which include (1) The patterns of the dietary habits differed between older adults with prefrailty or frailty and healthy individuals, depending on the diagnostic criteria of frailty that was used. (2) The problem in dietary habits appears from the stage of prefrailty diagnosed by the J-CHS. By contrast, it is not clear at the stage of prefrailty by the KCL. (3) The pattern of dietary habits in older adults with frailty or prefrailty varied significantly between men and women. The nutritional problem was more remarkable in frail or prefrail men. (4) Significant associations of the intakes of soluble dietary fiber, potassium, folate, and vitamin C with a frail status were observed in men, and an association of the intake of vitamin C with a frail status in women, according to the J-CHS.

Frailty represents a complex of clinical syndrome, and is characterized by decreased physiological reserve and increased vulnerability to stressors, resulting from multiple impairments across different organs [1,2]. Frailty is caused by multiple factors, including aging-associated loss of skeletal muscle mass, reduced nutritional intake, and low physical activity [1,2]. In recent times, the neuropsychiatric status, including cognitive impairment and depression, as well as social conditions such as solitude, have been shown to contribute to frailty [1,2,21]. Therefore, numerous tools to measure the frailty status have been developed. These can be divided into two groups: the phenotype model and the accumulation of deficits model, as represented by the CHS criteria, and the Frailty index, respectively [21]. In this study, we used the J-CHS as the phenotype model, and the KCL as the accumulation of deficits model. The KCL was originally designed to screen Japanese community-dwelling elderly adults whose status could potentially progress to the point of requiring long-term care [16]. Satake et al. evaluated the adequacy of a diagnosis of frailty severity based on the KCL criteria or the CHS criteria and reported that the respective sensitivity and specificity of these methods were 70.3% and 78.3% for prefrail individuals and 89.5% and 80.7% for frail individuals [16]. In this study, the two frailty diagnoses had a concordance rate of approximately 60%. As the KCL assesses the seven subdomains, including instrumental ADL, physical function, nutrition, oral function, socialization, memory, and mood [11,16], the impact of the physical aspect of frailty may become diluted by the domains of cognitive impairment, depression, and social frailty in individuals identified as frail using this model. We speculate that recent nutritional issues may more strongly affect the physical aspect of frailty than the neuropsychiatric and social aspects. Accordingly, the differences in basic characteristics and dietary habits between the robust and prefrail or frail groups, as identified using the two distinct sets of criteria, seemed acceptable. Our results suggest that nutritional interventions should be customized according to the characteristics of the diagnostic tool. Furthermore, uniform nutritional instructions, such as increases in the calorie and protein intakes, might be redundant for the treatment and prevention of frailty and sarcopenia in older adults. Although the KCL includes two queries about the nutritional status and three about oral function [11,16], the total score is used to diagnose frailty. Therefore, the J-CHS might more sensitively detect abnormal dietary habits in an earlier stage of the frailty cycle. Nutritional items should be extracted from the KCL prior to reevaluation, given the risk that this comprehensive frailty rating system might underestimate issues with a patient’s dietary habits. Accordingly, in a recent KCL-based study, the authors evaluated the associations between the frequency of protein-rich food intake and the frailty subdomains of KCL [11]. Further studies are needed to conclude which frailty assessment tool can better detect potential nutritional disturbances in community-dwelling older adults.

In our study, the nutritional issue had a more remarkable effect in frail men, compared with frail women. Notably, the male subjects included higher numbers of current smokers and alcohol drinkers, compared with the female subjects (Table 1). The excess intake of these luxury grocery items might affect the nutritional statuses of men, consistent with previous studies demonstrating that alcohol consumption can limit nutrient absorption and bioavailability in the blood [22]. This finding suggests a need for increased daily living guidance concerning these luxury grocery items, especially for men. By contrast, women exhibited a significantly higher prevalence of sarcopenia and osteoporosis (Table 1). In other words, a sex-related discrepancy exists between the possible involvement of malnutrition in the development of frailty and the higher prevalence of some downstream hallmarks of frailty.

The population in this study appeared to be healthy and well-nourished and had a relatively high intake of protein (Figure 1 and Figure 2). In such a population, the effect of insufficient nutritional intake on the development of frailty may have been affected by other factors. Additionally, inflammation, oxidative stress, and neurohumoral factors may contribute to the development of frailty [23,24]. Our results led us to speculate that men were more vulnerable to nutritional issues, whereas comorbidities might have a relatively greater effect on the development of frailty in women. Given the nature of frailty as a complex clinical syndrome comprising low physical activity, exhaustion, body weight loss, and other factors, the nutritional issue identified in this study might be more responsible for these components of frailty, rather than a low skeletal muscle mass or low bone density. We note the metabolisms of both bone and muscle are closely associated with sex hormones [25]. Furthermore, the prevalence of osteoporosis and sarcopenia may be affected by comorbid orthopedic diseases (e.g., osteoarthritis of the knee and lumbar spondylosis), most of which develop independently of the nutritional status [26]. These factors may also be responsible for the sex-related discrepancy between the involvement of malnutrition in the development of frailty and the prevalence of some downstream hallmarks of frailty. Therefore, discussions of sex differences in the pathophysiology of frailty should distinguish the osteoarticular and muscular components of frailty from other components.

Although self-administered dietary assessment methods, including questionnaires, cannot assess energy intake properly [8,19], this study showed no difference in energy intake between robust, prefrail, and frail individuals in the three populations, except for men diagnosed by the KCL (Table 2, Table 3, Table 4 and Table 5). This finding suggests that lower energy intake is responsible for decreased intakes of macronutrients in frail men according to the KCL and the frailty might develop independently of total energy intake. Previous studies, focusing on protein intake in elderly adults demonstrated that 1.0 to 1.2 g/kg/day protein intake is minimal, and 1.3 to 1.5 g/kg/day protein intake is recommended for elderly adults to perform endurance or resistance training [25]. More recently, Japanese investigators demonstrated close association between the frequency of protein-rich food intake, such as seafood and dairy products, and the prevalence of frailty diagnosed by KCL [11]. Although malnutrition, including lower protein intake, was observed in frail men diagnosed by KCL, we could not find any close association between protein intake and the prevalence of frailty in other groups. In the present study, protein intake of the participants was higher, compared with the general nutrient intake recommendations [13,25]. This may be because our participants had high physical performance levels and a good nutritional status. However, the importance of adequate protein intake for the prevention of frailty cannot be discounted.

By contrast, lower intake of soluble dietary fiber, potassium, vitamin C, and folate was commonly observed in prefrail or frail older adults, compared with the healthy people among the two different diagnostic methods for the robust, prefrail, and frail men (Figure 1 and Figure 2, and Table 2 and Table 4). In elderly adults whose intake of total energy and protein is sufficient, the variety of food groups may become more important. Dietary fibers have gained renewed interest as they are degraded by enteric bacteria in the colon into short chain fatty acids, which play an important role in maintaining the health of the colon and the whole body [27,28]. Recent papers suggested that elevated dietary fiber intake is closely associated with the health of elderly adults as well as longevity [29]. Our results were consistent with a previous paper from the Osteoporotic Fractures in Men (MrOS) study in the United States of America, which demonstrated that a higher intake of dietary fiber, but not of protein, significantly decreased the risk of prefrailty or frailty in community-dwelling older men [30]. To prevent the progression of frailty in older adults with sufficient energy and protein intake, the significance of consuming more dietary fiber should be clarified in future studies. In the frailty cycle, chronic inflammation, increased oxidative stress, and/or imbalances in the neurohumoral factors may contribute to the development of frailty independent of the general nutritional status [23,24]. Therefore, previous investigations demonstrated the association between frailty and micronutrients, especially vitamins and carotenoids, that may have antioxidant or anti-inflammatory properties [9,15]. Among various minerals and vitamins, lower intake of vitamin C and folate was frequently found in observational studies, including our study [9,15]. These results strongly suggest that consuming fruits and green and yellow vegetables is desirable to prevent frailty, in addition to preventing atherosclerosis.

This study has some limitations. First, due to its cross-sectional design, it was impossible to describe causal relationships. Longitudinal and interventional studies are required to quantitatively evaluate food intakes and to clarify the causal relationships in the future. Second, the survey was conducted in individuals who were capable of visiting our hospital for health check-ups using transportation means. In addition, we performed the dietary survey using a self-administered questionnaire. Therefore, our participants were relatively healthy. In fact, the proportion of frail participants was lower in this study than in previous studies performed in Japan [16,18]. It is often observed that frailty in older adults is accompanied with mild cognitive impairment [2,21]. Exclusion of individuals who displayed impaired cognitive function (MMSE score less than 24) may contribute to the lower prevalence of frailty in this study. Fourth, due to the lower prevalence of frailty in this study, it is possible that the significance of specific macronutrients or food group intake in frail older adults might become obscure due to statistical limitations. Finally, we failed to ask the participants questions about their marital status and whether they live alone or not. Eating alone may contribute to the development of frailty, at least in part, through enhancing nutritional imbalances.

In conclusion, our results demonstrate the association between frailty and dietary habits in rural community-dwelling older adults. In addition, our study suggests that attention should be paid to the diagnostic criteria of frailty and differences between sexes when nutritional interventions for older adults with frailty are planned. To prevent the progression of frailty in older adults who have sufficient energy and protein intake, a comprehensive nutritional management plan, including optimal intake of dietary fiber, minerals, and vitamins, should be developed in a longitudinal study in the near future.

## Figures and Tables

**Figure 1 nutrients-10-01982-f001:**
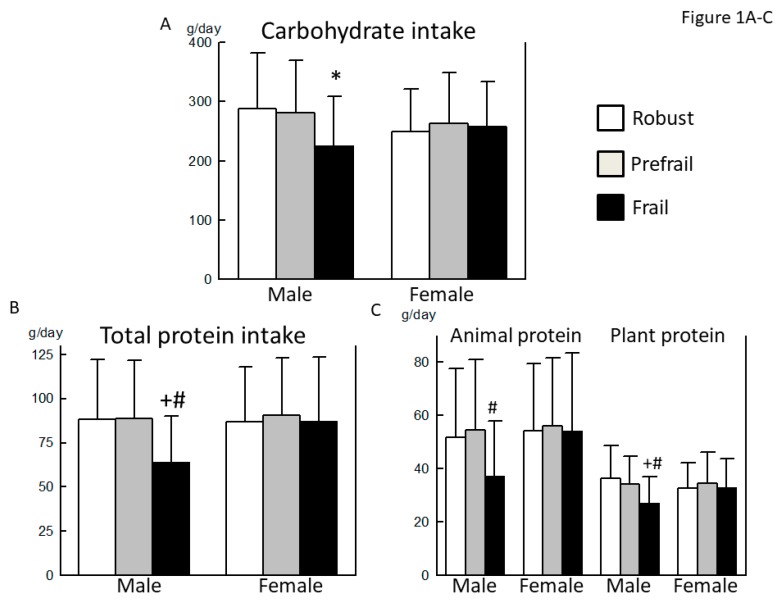
Comparison of macronutrient intake between robust, prefrail, and frail individuals, diagnosed by the Kihon checklist. (**A**): Carbohydrate intake, (**B**): Total protein intake, (**C**): Animal and plant protein intake, (**D**): Total fat intake, (**E**): Animal and plant fat intake, and (**F**): Total, soluble, and insoluble dietary fiber intake. ∗: *p* < 0.05 vs. Robust, +: *p* < 0.01 vs. Robust, #: *p* < 0.05 vs. Prefrail, &: *p* < 0.01 vs. Prefrail.

**Figure 2 nutrients-10-01982-f002:**
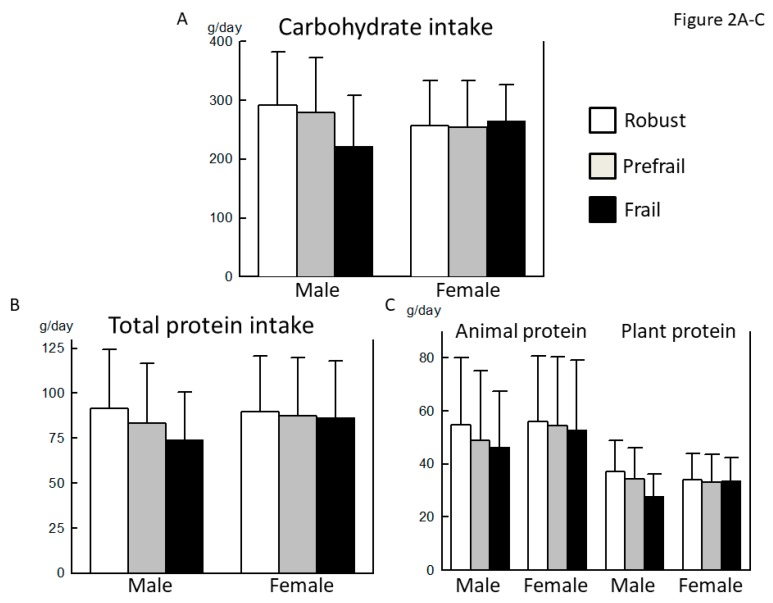
Comparison of macronutrient intake between robust, prefrail, and frail individuals, diagnosed by the Japanese version of the Cardiovascular Health Study. (**A**): Carbohydrate intake, (**B**): Total protein intake, (**C**): Animal and plant protein intake, (**D**): Total fat intake, (**E**): Animal and plant fat intake, and (**F**): Total, soluble, and insoluble dietary fiber intake. ∗: *p* < 0.05 vs. Robust.

**Table 1 nutrients-10-01982-t001:** Basic characteristics and the diagnosis of frailty in the 800 participants.

	Total (*n* = 800)	Male (*n* = 254)	Female (*n* = 546)	*p*-Value
Age (years)	72.6 ± 5.8	73.1 ± 6.2	72.4 ± 5.6	n.s.
Height (cm)	155.5 ± 8.1	164.1 ± 5.8	151.5 ± 5.6	<0.001
BW (kg)	54.9 ± 9.4	62.7 ± 9.0	51.3 ± 7.1	<0.001
BMI (kg/m^2^)	22.6 ± 2.9	23.2 ± 2.9	22.3 ± 2.9	<0.001
SBP (mmHg)	138.9 ± 17.6	137.5 ± 18.2	139.6 ± 17.4	n.s.
DBP (mmHg)	80.5 ± 10.7	80.3 ± 11.4	80.6 ± 10.3	n.s.
MMSE score	28.3 ± 1.8	28.2 ± 1.8	28.4 ± 1.8	n.s.
Smoking habit				
Never	579 (72.4)	71 (28.0)	508 (93.0)	
Past	181 (22.6)	155 (61.0)	26 (4.8)	
Current	40 (5.0)	28 (11.0)	12 (2.2)	<0.001
Alcohol drinking	349 (43.6)	192 (75.6)	157 (28.8)	<0.001
Exercise habit	508 (63.5)	154 (60.6)	354 (64.8)	n.s.
Comorbidities				
Hypertension	362 (45.3)	128 (50.4)	234 (42.9)	0.046
DM	98 (12.3)	50 (19.7)	48 (8.8)	<0.001
Dyslipidemia	192 (24.0)	44 (17.3)	148 (27.1)	0.003
Chronic liver disease	34 (4.3)	16 (6.3)	18 (3.3)	n.s.
CKD	27 (3.4)	11 (4.3)	16 (2.9)	n.s.
CVD	70 (8.8)	34 (13.4)	36 (6.6)	0.002
Gastrointestinal disease	48 (6.0)	18 (7.1)	30 (5.5)	n.s.
Asthma and COPD	24 (3.0)	7 (2.8)	17 (3.1)	n.s.
Thyroid disease	35 (4.4)	8 (3.1)	27 (4.9)	n.s.
Osteoporosis	97 (12.1)	3 (1.2)	94 (17.2)	<0.001
RA and collagen disease	17 (2.1)	4 (1.6)	13 (2.4)	n.s.
History of cancer	58 (7.3)	30 (11.8)	28 (4.2)	<0.001
Frailty diagnosis by the KCL criteria	(*n* = 800)	(*n* = 254)	(*n* = 546)	
Robust	476 (59.5)	163 (64.3)	313 (57.3)	
Prefrail	257 (32.1)	72 (28.2)	185 (33.9)	
Frail	67 (8.4)	19 (7.5)	48 (8.8)	n.s.
Frailty diagnosis by the J-CHS criteria	(*n* = 796)	(*n* = 252)	(*n* = 544)	
Robust	331 (41.6)	104 (41.3)	227 (41.7)	
Prefrail	436 (54.8)	139 (55.2)	294 (54.0)	
Frail	32 (4.0)	9 (3.6)	23 (4.2)	n.s.
Sarcopenia diagnosis by the AWGS criteria	(*n* = 788)	(*n* = 250)	(*n* = 538)	
Robust	542 (68.8)	182 (72.8)	360 (66.9)	
Presarcopenia	217 (27.5)	65 (26.0)	152 (28.3)	
Sarcopenia	29 (3.7)	3 (1.2)	26 (4.8)	0.026

The data in the table represent the prevalence of a condition by the total number of individuals (mean ± SD or number), with % of population in parenthesis next to it. This applies to all variables unless specific units are provided. BW: body weight, BMI: body mass index, SBP: systolic blood pressure, DBP: diastolic blood pressure, MMSE: Mini-Mental State Examination, DM: diabetes mellitus, CKD: chronic kidney disease, CVD: cardiovascular disease, COPD: cardiopulmonary disease, RA: rheumatoid arthritis, KCL: Kihon Checklist, J-CHS: Japanese version of the Cardiovascular Health Study, AWGS: Asian Working Group for Sarcopenia.

**Table 2 nutrients-10-01982-t002:** Basic characteristics and dietary habits among frail, prefrail, and robust men, as diagnosed by the KCL criteria.

	Robust (*n* = 163)	Prefrail (*n* = 72)	Frail (*n* = 19)	*p*-Value
Age (years)	72.4 ± 5.6	74.7 ± 7.0	73.3 ± 6.9	0.026 (R vs. P)
Height (cm)	164.3 ± 5.8	163.6 ± 6.4	163.7 ± 4.0	n.s.
BW (kg)	63.2 ± 8.5	62.1 ± 10.1	60.0 ± 8.2	n.s.
BMI (kg/m^2^)	23.4 ± 2.8	23.1 ± 3.2	22.4 ± 3.1	n.s.
SBP (mmHg)	137.5 ± 19.0	136.9 ± 15.9	138.7 ± 19.8	n.s.
DBP (mmHg)	81.4 ± 11.6	77.7 ± 10.6	80.9 ± 11.3	n.s.
MMSE score	28.3 ± 1.8	28.0 ± 1.9	28.1 ± 1.9	n.s.
Smoking Habit				
Never	48 (29.4)	18 (25.0)	5 (26.3)	
Past	98 (60.1)	49 (68.1)	8 (42.1)	
Current	17 (10.4)	5 (6.9)	6 (31.6)	0.033
Alcohol drinking	126 (77.3)	51 (70.8)	15 (78.9)	n.s.
Exercise habit	106 (65.0)	39 (54.2)	9 (47.4)	n.s.
Comorbidities				
Hypertension	81 (49.7)	38 (52.8)	9 (47.4)	n.s.
DM	32 (19.6)	17 (23.6)	1 (5.3)	n.s.
Dyslipidemia	26 (16.0)	17 (23.6)	1 (5.3)	n.s.
Chronic liver disease	10 (6.1)	4 (5.6)	2 (10.5)	n.s.
CKD	9 (5.5)	1 (1.4)	1 (5.3)	n.s.
CVD	20 (12.3)	13 (18.1)	1 (5.3)	n.s.
Gastrointestinal disease	13 (8.0)	4 (5.6)	1 (5.3)	n.s.
Asthma/ COPD	5 (3.1)	1 (1.4)	1 (5.3)	n.s.
Thyroid disease	6 (3.7)	2 (2.8)	0	n.s.
Osteoporosis	1 (0.6)	2 (2.8)	0	
RA and collagen disease	2 (1.2)	2 (2.8)	0	
History of cancer	21 (12.9)	7 (9.7)	2 (10.5)	n.s.
	Robust (*n* = 163)	Prefrail (*n* = 72)	Frail (*n* = 19)	*p*-value
Energy intake (kcal/day)	2250 ± 633	2239 ± 623	1807 ± 575	0.021 (R vs. F) 0.048 (P vs. F)
Energy intake/BW (kcal/day/kg)	36.2 ± 11.2	36.9 ± 11.9	30.6 ± 10.1	n.s.
Macro- and micronutrients				
Carbohydrate (% energy/day)	51.4 ± 8.3	50.3 ± 8.1	49.8 ± 8.5	n.s.
Total protein (% energy/day)	15.5 ± 3.0	15.8 ± 3.6	14.4 ± 3.8	n.s.
Fat (% energy/day)	25.2 ± 5.0	26.7 ± 4.8	24.6 ± 8.0	n.s.
Sodium (g/day)	5.14 ± 1.74	5.18 ± 1.63	4.05 ± 1.26	0.021 (R vs. F) 0.026 (P vs. F)
Potassium (g/day)	3.32 ± 1.32	3.18 ± 1.22	2.37 ± 0.98	0.007 (R vs. F) 0.039 (F vs. F)
Calcium (mg/day)	780 ± 365	768 ± 314	502 ± 245	0.003 (R vs. F) 0.009 (P vs. F)
Magnesium (mg/day)	332 ± 124	316 ± 109	239 ± 87	0.003 (R vs. F) 0.031 (P vs. F)
Phosphorus (mg/day)	1388 ± 555	1377 ± 500	980 ± 387	0.005 (R vs. F) 0.011 (P vs. F)
Iron (mg/day)	10.2 ± 4.1	9.8 ± 3.8	7.2 ± 3.1	0.005 (R vs. F) 0.033 (P vs. F)
Zinc (mg/day)	9.98 ± 3.39	9.88 ± 3.43	7.53 ± 2.77	0.008 (R vs. F) 0.019 (P vs. F)
Manganese (mg/day)	3.82 ± 1.32	3.49 ± 1.28	2.93 ± 1.14	n.s.
Copper (mg/day)	1.43 ± 0.49	1.34 ± 0.45	1.07 ± 0.44	0.007 (R vs. F)
Retinol equivalent (μg /day)	1104 ± 867	1099 ± 833	792 ± 651	n.s.
α-Carotene (μg/day)	507 ± 402	437 ± 373	333 ± 262	n.s.
β-Carotene (equivalent μg/day)	5098 ± 3113	4507 ± 2948	3043 ± 1973	0.014 (R vs. F)
Vitamin D (μg/day)	22.3 ± 16.5	21.8 ± 15.5	13.6 ± 11.1	n.s.
α-Tocopherol (mg/day)	9.32 ± 3.61	9.27 ± 3.45	6.53 ± 3.06	0.004 (R vs. F) 0.008 (P vs. F)
γ-Tocopherol (mg/day)	15.0 ± 5.4	15.4 ± 5.4	12.1 ± 6.1	0.049 (P vs. F)
Vitamin K (mg/day)	430 ± 239	369 ± 208	295 ± 176	0.038 (R vs. F)
Vitamin B1 (mg/day)	0.95 ± 0.36	0.95 ± 0.35	0.68 ± 0.29	0.007 (R vs. F) 0.012 (P vs. F)
Vitamin B2 (mg/day)	1.69 ± 0.65	1.67 ± 0.64	1.23 ± 0.50	0.009 (R vs. F) 0.021 (P vs. F)
Niacin (mg/day)	21.0 ± 8.3	21.1 ± 8.6	16.5 ± 7.2	n.s.
Vitamin B6 (mg/day)	1.64 ± 0.62	1.59 ± 0.60	1.26 ± 0.53	0.028 (R vs. F)
Vitamin B12 (μg/day)	13.7 ± 8.7	13.9 ± 8.5	9.9 ± 7.2	n.s.
Folate (μg/day)	471 ± 205	425 ± 188	329 ± 151	0.009 (R vs. F)
Pantothenic acid (mg/day)	8.22 ± 2.99	8.07 ± 2.93	6.09 ± 2.53	0.009 (R vs. F) 0.026 (P vs. F)
Vitamin C (mg/day)	161 ± 83	144 ± 73	104 ± 57	0.008 (R vs. F)
Cryptoxanthin (μg/day)	427 ± 402	405 ± 358	199 ± 192	0.036 (R vs. F)
Food group (g/day)				
Cereals	441 ± 170	424 ± 186	364 ± 161	n.s.
Potatoes	64.1 ± 60.6	60.2 ± 57.4	56.0 ± 50.3	n.s.
Sugar and sweeteners	6.01 ± 4.10	7.20 ± 5.43	4.28 ± 4.55	0.036 (P vs. F)
Soy products	90.8 ± 56.5	81.1 ± 50.5	62.3 ± 47.9	n.s.
Total vegetables	355 ± 204	322 ± 186	245 ± 142	n.s.
Fruits	159 ± 137	152 ± 119	119 ± 83	0.041 (R vs. F)
Fish and shellfish	114 ± 77	115 ± 79	79 ± 77	n.s.
Meats	72.1 ± 41.5	81.7 ± 54.7	66.1 ± 52.9	n.s.
Eggs	49.0 ± 31.8	51.1 ± 36.3	32.2 ± 26.2	n.s.
Dairy products	174 ± 126	199 ± 146	115 ± 141	0.039 (P vs. F)
Oil	12.0 ± 5.3	12.6 ± 5.8	10.3 ± 6.9	n.s.
Confectioneries	64.9 ± 53.9	62.0 ± 48.9	42.8 ± 49.8	n.s.

Mean ± SD or number (%), n.s.: not significant, R: Robust, P: Prefrail, F: Frail.

**Table 3 nutrients-10-01982-t003:** Basic characteristics and dietary habits among frail, prefrail, and robust women, as diagnosed by the KCL criteria.

	Robust (*n* = 313)	Prefrail (*n* = 185)	Frail (*n* = 48)	*p*-Value
Age (years)	71.7 ± 5.5	72.9 ± 5.6	75.4 ± 5.7	<0.001 (R vs. F) 0.042 (R vs. P) 0.018 (P vs. F)
Height (cm)	152.1 ± 5.3	151.2 ± 5.7	148.5 ± 6.2	<0.001 (R vs. F) 0.009 (P vs. F)
BW (kg)	51.8 ± 7.0	50.6 ± 7.1	50.4 ± 7.6	n.s.
BMI (kg/m^2^)	22.4 ± 2.7	22.2 ± 3.0	22.8 ± 3.5	n.s.
SBP (mmHg)	139.5 ± 16.6	139.1 ± 18.5	142.3 ± 17.9	n.s.
DBP (mmHg)	80.8 ± 10.1	80.4 ± 10.9	80.6 ± 9.6	n.s.
MMSE score	28.5 ± 1.8	28.4 ± 1.8	28.0 ± 1.8	n.s.
Smoking Habit Never	293 (93.6)	170 (91.9)	45 (93.8)	
Past	13 (4.2)	11 (5.9)	2 (4.2)	
Current	7 (2.2)	4 (2.2)	1 (2.1)	n.s.
Alcohol drinking	99 (31.6)	52 (28.1)	6 (12.5)	0.024
Exercise habit	215 (68.7)	119 (64.3)	20 (41.7)	0.001
Comorbidities				
Hypertension	125 (39.9)	84 (45.4)	25 (52.1)	n.s.
DM	24 (7.7)	17 (9.2)	7 (14.6)	n.s.
Dyslipidemia	88 (28.1)	49 (26.5)	11 (22.9)	n.s.
Chronic liver disease	11 (3.5)	4 (2.2)	3 (6.3)	n.s.
CKD	6 (1.9)	5 (2.7)	5 (10.4)	0.005
CVD	18 (5.8)	16 (8.6)	4 (8.3)	n.s.
Gastrointestinal disease	13 (4.2)	9 (4.9)	8 (16.7)	0.002
Asthma/ COPD	4 (1.3)	10 (5.4)	3 (6.3)	0.016
Thyroid disease	13 (4.2)	13 (7.0)	1 (2.1)	n.s.
Osteoporosis	42 (13.4)	40 (21.6)	12 (25.0)	0.021
RA and collagen disease	9 (2.9)	4 (2.2)	0	
History of cancer	12 (3.8)	10 (5.4)	6 (12.5)	0.039
	Robust (*n* = 313)	Prefrail (*n* = 185)	Frail (*n* = 48)	*p*-value
Energy intake (kcal/day)	1928 ± 537	2032 ± 594	1994 ± 624	0.031 (R vs. P)
Energy intake/BW (kcal/day/kg)	38.0 ± 12.3	40.9 ± 13.4	40.4 ± 13.6	n.s.
Macro- and micronutrients				
Carbohydrate (% energy/day)	51.9 ± 7.0	51.9 ± 7.7	52.2 ± 6.7	n.s.
Total protein (% energy/day)	18.0 ± 3.3	17.8 ± 3.5	17.2 ± 3.1	n.s.
Fat (% energy/day)	28.7 ± 4.8	28.6 ± 5.2	29.5 ± 4.8	n.s.
Sodium (g/day)	4.66 ± 1.53	4.92 ± 1.56	4.71 ± 1.77	n.s.
Potassium (g/day)	3.41 ± 1.14	3.52 ± 1.29	3.33 ± 1.27	n.s.
Calcium (mg/day)	790 ± 291	827 ± 336	737 ± 272	n.s.
Magnesium (mg/day)	321 ± 105	334 ± 115	311 ± 114	n.s.
Phosphorus (mg/day)	1364 ± 483	1422 ± 519	1340 ± 542	n.s.
Iron (mg/day)	10.2 ± 3.5	10.5 ± 3.6	10.1 ± 3.8	n.s.
Zinc (mg/day)	9.56 ± 3.07	9.90 ± 3.08	9.58 ± 3.55	n.s.
Manganese (mg/day)	3.86 ± 1.11	3.92 ± 1.22	3.70 ± 1.20	n.s.
Copper (mg/day)	1.34 ± 0.40	1.39 ± 0.45	1.35 ± 0.47	n.s.
Retinol equivalent (μg /day)	1019 ± 688	1018 ± 586	934 ± 478	n.s.
α-Carotene (μg/day)	649 ± 423	689 ± 490	592 ± 399	n.s.
β-Carotene equivalent (μg/day)	5820 ± 3188	6076 ± 3433	5558 ± 3201	n.s.
Vitamin D (μg/day)	24.1 ± 15.9	25.7 ± 16.0	22.2 ± 16.1	n.s.
α-Tocopherol (mg/day)	9.44 ± 3.37	9.93 ± 3.69	9.86 ± 4.16	n.s.
γ-Tocopherol (mg/day)	13.6 ± 4.8	14.7 ± 5.7	14.7 ± 5.3	n.s.
Vitamin K (mg/day)	413 ± 202	423 ± 216	399 ± 232	n.s.
Vitamin B1 (mg/day)	0.98 ± 0.33	1.01 ± 0.35	0.97 ± 0.38	n.s.
Vitamin B2 (mg/day)	1.70 ± 0.54	1.73 ± 0.59	1.66 ± 0.57	n.s.
Niacin (mg/day)	21.3 ± 8.5	21.9 ± 8.2	20.9 ± 9.8	n.s.
Vitamin B6 (mg/day)	1.63 ± 0.59	1.67 ± 0.61	1.62 ± 0.68	n.s.
Vitamin B12 (μg/day)	13.7 ± 8.1	14.4 ± 8.2	13.3 ± 9.1	n.s.
Folate (μg/day)	479 ± 178	484 ± 188	454 ± 185	n.s.
Pantothenic acid (mg/day)	8.00 ± 2.59	8.20 ± 2.83	7.99 ± 2.97	n.s.
Vitamin C (mg/day)	182 ± 75	183 ± 84	175 ± 78	n.s.
Cryptoxanthin (μg/day)	489 ± 352	519 ± 437	454 ± 331	n.s.
Food group (g/day)				
Cereals	336 ± 142	356 ± 147	348 ± 139	n.s.
Potatoes	68.9 ± 53.7	73.0 ± 62.2	74.3 ± 62.9	n.s.
Sugar and sweeteners	5.90 ± 4.15	6.39 ± 5.03	6.48 ± 4.55	n.s.
Soy products	88.0 ± 45.8	91.6 ± 54.1	84.0 ± 46.3	n.s.
Total vegetables	377 ± 186	392 ± 201	365 ± 184	n.s.
Fruits	180 ± 114	180 ± 135	178 ± 116	n.s.
Fish and shellfish	120 ± 76	128 ± 79	120 ± 100	n.s.
Meats	79.9 ± 51.5	77.3 ± 39.2	81.6 ± 60.6	n.s.
Eggs	43.9 ± 26.3	44.8 ± 31.0	54.0 ± 35.6	n.s.
Dairy products	174 ± 97	179 ± 119	159 ± 88	n.s.
Oil	10.0 ± 5.0	11.0 ± 5.4	12.0 ± 5.4	0.036 (R vs. F)
Confectioneries	67.7 ± 50.2	73.3 ± 63.4	76.6 ± 49.0	n.s.

Mean ± SD or number (%), n.s.: not significant, R: Robust, P: Prefrail, F: Frail.

**Table 4 nutrients-10-01982-t004:** Basic characteristics and dietary habits among frail, prefrail, and robust men, as diagnosed by the J-CHS criteria.

	Robust (*n* = 104)	Prefrail (*n* = 139)	Frail (*n* = 9)	*p*-Value
Age (years)	73.4 ± 5.0	72.8 ± 7.0	74.9 ± 7.0	n.s.
Height (cm)	164.4 ± 6.0	163.8 ± 5.8	165.6 ± 5.4	n.s.
BW (kg)	62.6 ± 8.7	62.8 ± 9.1	61.4 ± 11.0	n.s.
BMI (kg/m^2^)	23.2 ± 2.9	23.4 ± 2.9	22.4 ± 4.2	n.s.
SBP (mmHg)	136.3 ± 16.5	137.8 ± 19.6	142.6 ± 14.0	n.s.
DBP (mmHg)	78.7 ± 10.9	81.3 ± 11.8	81.9 ± 10.0	n.s.
MMSE score	28.2 ± 1.9	28.1 ± 1.9	28.3 ± 1.8	n.s.
Smoking Habit				
Never	30 (28.8)	39 (28.3)	2 (22.2)	
Past	65 (62.5)	83 (59.7)	5 (55.6)	
Current	9 (8.7)	17 (12.2)	2 (22.2)	n.s.
Alcohol drinking	76 (73.1)	108 (77.7)	7 (77.8)	n.s.
Exercise habit	104 (100.0)	50 (75.2)	0	
Comorbidities				
Hypertension	45 (43.3)	77 (55.4)	5 (55.6)	n.s.
DM	26 (25.0)	20 (14.4)	3 (33.3)	n.s.
Dyslipidemia	19 (18.3)	24 (17.3)	0	
Chronic liver disease	9 (8.7)	7 (5.0)	0	
CKD	6 (5.8)	4 (2.9)	1 (11.1)	n.s.
CVD	17 (16.3)	17 (12.2)	0	
Gastrointestinal disease	9 (8.7)	9 (6.5)	0	
Asthma/ COPD	4 (3.8)	2 (1.4)	1 (11.1)	n.s.
Thyroid disease	4 (3.8)	4 (2.9)	0	
Osteoporosis	1 (1.0)	1 (0.7)	1 (11.1)	0.02
RA and collagen disease	1 (1.0)	3 (2.2)	0	
History of cancer	14 (13.5)	14 (10.1)	2 (22.2)	n.s.
	Robust (*n* = 104)	Prefrail (*n* = 139)	Frail (*n* = 9)	*p*-value
Energy intake (kcal/day)	2305 ± 605	2170 ± 644	1809 ± 567	n.s.
Energy intake/BW (kcal/day/kg)	37.4 ± 11.0	35.2 ± 11.5	30.6 ± 11.6	n.s.
Macro- and micronutrients				
Carbohydrate (% energy/day)	50.7 ± 7.9	51.6 ± 8.4	48.4 ± 9.2	n.s.
Total protein (% energy/day)	15.8 ± 3.0	15.2 ± 3.3	16.6 ± 4.4	n.s.
Fat (% energy/day)	25.9 ± 4.6	25.1 ± 5.6	28.0 ± 6.7	n.s.
Sodium (g/day)	5.26 ± 1.63	4.93 ± 1.75	4.71 ± 1.44	n.s.
Potassium (g/day)	3.47 ± 1.31	3.04 ± 1.25	2.64 ± 1.29	0.029 (R vs. P)
Calcium (mg/day)	814 ± 354	723 ± 344	560 ± 325	n.s.
Magnesium (mg/day)	342 ± 123	308 ± 116	265 ± 95	n.s.
Phosphorus (mg/day)	1440 ± 535	1300 ± 532	1156 ± 472	n.s.
Iron (mg/day)	10.68 ± 4.03	9.32 ± 3.44	8.15 ± 2.92	0.023 (R vs. P)
Zinc (mg/day)	10.29 ± 3.36	9.46 ± 3.44	8.53 ± 2.68	n.s.
Manganese (mg/day)	3.83 ± 1.29	3.60 ± 1.34	2.78 ± 0.90	n.s.
Copper (mg/day)	1.46 ± 0.50	1.33 ± 0.47	1.15 ± 0.41	n.s.
Retinol equivalent (μg /day)	1210 ± 1028	984 ± 665	1094 ± 946	n.s.
α-Carotene (μg/day)	530 ± 400	440 ± 371	420 ± 479	n.s.
β-Carotene equivalent (μg/day)	5405 ± 3113	4401 ± 2930	3742 ± 3093	0.029 (R vs. P)
Vitamin D (μg/day)	22.9 ± 16.4	20.3 ± 15.3	19.0 ± 14.6	n.s.
α-Tocopherol (mg/day)	9.69 ± 3.41	8.69 ± 3.69	7.80 ± 2.80	n.s.
γ-Tocopherol (mg/day)	15.2 ± 5.2	14.7 ± 5.7	13.9 ± 4.6	n.s.
Vitamin K (mg/day)	443 ± 238	380 ± 224	313 ± 140	n.s.
Vitamin B1 (mg/day)	1.00 ± 0.35	0.88 ± 0.36	0.81 ± 0.38	0.024 (R vs. P)
Vitamin B2 (mg/day)	1.78 ± 0.65	1.57 ± 0.63	1.33 ± 0.56	0.032 (R vs. P)
Niacin (mg/day)	22.0 ± 8.5	19.8 ± 8.3	17.7 ± 6.5	n.s.
Vitamin B6 (mg/day)	1.72 ± 0.61	1.52 ± 0.60	1.27 ± 0.60	0.026 (R vs. P)
Vitamin B12 (μg/day)	14.4 ± 8.8	12.7 ± 8.3	11.5 ± 6.5	n.s.
Folate (μg/day)	493 ± 208	419 ± 190	365 ± 175	0.011 (R vs. P)
Pantothenic acid (mg/day)	8.60 ± 3.01	7.67 ± 2.93	6.80 ± 2.61	0.043 (R vs. P)
Vitamin C (mg/day)	171 ± 82	140 ± 76	118 ± 90	0.008 (R vs. P)
Cryptoxanthin (μg/day)	484 ± 429	358 ± 325	358 ± 455	n.s.
Food group (g/day)				
Cereals	427 ± 166	438 ± 181	381 ± 178	n.s.
Potatoes	70.2 ± 61.5	56.7 ± 54.9	65.0 ± 82.7	n.s.
Sugar and sweeteners	6.47 ± 4.20	5.93 ± 4.62	6.95 ± 7.11	n.s.
Soy products	93.1 ± 59.9	81.4 ± 48.9	61.0 ± 34.0	n.s.
Total vegetables	369 ± 194	317 ± 197	281 ± 206	n.s.
Fruits	172 ± 150	137 ± 109	107 ± 104	n.s.
Fish and shellfish	118 ± 78	106 ± 74	96 ± 60	n.s.
Meats	80.1 ± 48.1	70.1 ± 46.3	76.7 ± 30.5	n.s.
Eggs	51.2 ± 29.7	45.9 ± 33.8	45.0 ± 35.1	n.s.
Dairy products	188 ± 122	174 ± 144	122 ± 101	n.s.
Oil	12.1 ± 5.4	12.0 ± 5.7	11.7 ± 5.3	n.s.
Confectioneries	70.7 ± 56.1	59.0 ± 49.9	30.7 ± 20.3	0.038 (R vs. F)

Mean ± SD or number (%), n.s.: not significant, R: Robust, P: Prefrail, F: Frail.

**Table 5 nutrients-10-01982-t005:** Basic characteristics and dietary habits among frail, prefrail, and robust women, as diagnosed by the J-CHS criteria.

	Robust (*n* = 227)	Prefrail (*n* = 294)	Frail (*n* = 23)	*p*-Value
Age (years)	72.2 ± 5.4	72.3 ± 5.5	76.5 ± 7.8	0.002 (R vs. F) 0.002 (P vs. F)
Height (cm)	152.2 ± 5.2	151.2 ± 5.4	147.6 ± 7.6	<0.001(R vs. F) 0.007 (P vs. F)
BW (kg)	51.0 ± 6.4	51.7 ± 7.5	49.1 ± 8.9	n.s.
BMI (kg/m^2^)	22.0 ± 2.5	22.6 ± 3.0	22.5 ± 4.2	0.042 (R vs. P)
SBP (mmHg)	138.8 ± 17.3	140.0 ± 17.4	142.6 ± 17.8	n.s.
DBP (mmHg)	80.5 ± 10.7	80.8 ± 10.2	80.4 ± 9.2	n.s.
MMSE score	28.4 ± 1.7	28.4 ± 1.8	27.6 ± 1.9	n.s.
Smoking Habit				
Never	215 (94.7)	269 (91.5)	22 (95.7)	
Past	7 (3.1)	19 (6.5)	0	
Current	5 (2.2)	6 (2.0)	1 (4.3)	n.s.
Alcohol drinking	65 (28.6)	89 (30.3)	2 (8.7)	n.s.
Exercise habit	227 (100.0)	121 (41.2)	3 (13.0)	<0.001
Comorbidities				
Hypertension	90 (39.6)	129 (43.9)	14 (60.9)	n.s.
DM	20 (8.8)	24 (8.2)	4 (17.4)	n.s.
Dyslipidemia	50 (22.0)	94 (32.0)	3 (13.0)	0.012
Chronic liver disease	5 (2.2)	10 (3.4)	3 (13.0)	0.021
CKD	1 (0.4)	12 (4.1)	2 (8.7)	0.009
CVD	11 (4.8)	23 (7.8)	2 (8.7)	n.s.
Gastrointestinal disease	8 (3.5)	18 (6.1)	4 (17.4)	0.017
Asthma/ COPD	4 (1.8)	11 (3.7)	2 (8.7)	n.s.
Thyroid disease	8 (3.5)	17 (5.8)	2 (8.7)	n.s.
Osteoporosis	29 (12.8)	60 (20.4)	4 (17.4)	n.s.
RA and collagen disease	7 (3.1)	4 (1.4)	1 (4.3)	n.s.
History of cancer	10 (4.4)	15 (5.1)	3 (13.0)	n.s.
	Robust (*n* = 227)	Prefrail (*n* = 294)	Frail (*n* = 23)	*p*-value
Energy intake (kcal/day)	1978 ± 551	1962 ± 585	2010 ± 472	n.s.
Energy intake/BW (kcal/day/kg)	39.5 ± 12.6	38.7 ± 13.0	41.9 ± 12.3	n.s.
Macro- and micronutrients				
Carbohydrate (% energy/day)	52.0 ± 7.0	51.8 ± 7.5	53.3 ± 5.9	n.s.
Total protein (% energy/day)	18.1 ± 3.3	17.7 ± 3.4	16.8 ± 3.2	n.s.
Fat (% energy/day)	28.6 ± 4.8.	28.8 ± 5.1	29.0 ± 4.5	n.s.
Sodium (g/day)	4.78 ± 1.53	4.73 ± 1.61	4.72 ± 1.57	n.s.
Potassium (g/day)	3.57 ± 1.16	3.33 ± 1.21	3.47 ± 1.42	0.048 (R vs. P)
Calcium (mg/day)	826 ± 302	776 ± 301	819 ± 381	n.s.
Magnesium (mg/day)	334 ± 106	317 ± 110	325 ± 123	n.s.
Phosphorus (mg/day)	1414 ± 492	1357 ± 504	1379 ± 529	n.s.
Iron (mg/day)	10.6 ± 3.5	10.1 ± 3.6	10.4 ± 3.7	n.s.
Zinc (mg/day)	9,78 ± 3.11	9.61 ± 3.16	9.56 ± 3.07	n.s.
Manganese (mg/day)	3.92 ± 1.12	3.83 ± 1.18	3.87 ± 1.14	n.s.
Copper (mg/day)	1.38 ± 0.42	1.33 ± 0.43	1.41 ± 0.45	n.s.
Retinol equivalent (μg /day)	1080 ± 732	958 ± 555	1016 ± 580	n.s.
α-Carotene (μg/day)	680 ± 423	633 ± 445	703 ± 565	n.s.
β-Carotene equivalent (μg/day)	6111 ± 3083	5657 ± 3236	6295 ± 4494	n.s.
Vitamin D (μg/day)	25.5 ± 16.2	23.9 ± 15.8	21.6 ± 15.1	n.s.
α-Tocopherol (mg/day)	9.80 ± 3.33	9.45 ± 3.60	10.52 ± 4.70	n.s.
γ-Tocopherol (mg/day)	14.0 ± 4.9	14.0 ± 5.4	15.6 ± 4.5	n.s.
Vitamin K (mg/day)	432 ± 203	401 ± 205	421 ± 280	n.s.
Vitamin B1 (mg/day)	1.01 ± 0.33	0.97 ± 0.35	0.95 ± 0.33	n.s.
Vitamin B2 (mg/day)	1.75 ± 0.54	1.67 ± 0.57	1.75 ± 0.60	n.s.
Niacin (mg/day)	22.0 ± 8.3	21.2 ± 8.7	19.9 ± 8.1	n.s.
Vitamin B6 (mg/day)	1.69 ± 0.58	1.61 ± 0.61	1.60 ± 0.63	n.s.
Vitamin B12 (μg/day)	14.1 ± 7.9	13.8 ± 8.4	13.1 ± 9.0	n.s.
Folate (μg/day)	499 ± 181	462 ± 176	480 ± 224	0.049 (R vs. P)
Pantothenic acid (mg/day)	8.31 ± 2.65	7.89 ± 2.73	8.08 ± 2.62	n.s.
Vitamin C (mg/day)	193 ± 78	173 ± 75	180 ± 103	0.013 (R vs. P)
Cryptoxanthin (μg/day)	527 ± 378	474 ± 380	477 ± 437	n.s.
Food group (g/day)				
Cereals	340 ± 137	345 ± 149	376 ± 134	n.s.
Potatoes	76.8 ± 60.4	65.7 ± 55.5	73.4 ± 47.4	n.s.
Sugar and sweeteners	6.26 ± 4.14	5.94 ± 4.78	6.87 ± 4.50	n.s.
Soy products	91.5 ± 47.3	87.0 ± 49.6	89.9 ± 52.8	n.s.
Total vegetables	391 ± 185	372 ± 185	385 ± 262	n.s.
Fruits	193 ± 123	170 ± 116	190 ± 166	n.s.
Fish and shellfish	124 ± 75	120 ± 80	123 ± 108	n.s.
Meats	79.5 ± 50.4	80.6 ± 48.4	58.7 ± 23.2	n.s.
Eggs	44.3 ± 26.0	44.5 ± 29.6	60.1 ± 42.4	0.036 (R vs. F) 0.033 (P vs. F).
Dairy products	179 ± 101	169 ± 104	211 ± 133	n.s.
Oil	10.4 ± 5.2	10.6 ± 5.2	11.7 ± 4.8	n.s.
Confectioneries	70.1 ± 53.8	71.8 ± 56.8	59.2 ± 39.5	n.s.

Mean ± SD or number (%), n.s.: not significant, R: Robust, P: Prefrail, F: Frail.

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
