# Peer review of "The Relationship between Dietary Habits and Frailty in Rural Japanese Community-Dwelling Older Adults: Cross-Sectional Observation Study Using a Brief Self-Administered Dietary History Questionnaire"

_nutrients, 2018, doi:10.3390/nu10121982_

Reviewer 1 Report

Dear Authors

the topic of the paper is relevant, but final results should be evaluated by using

multivariate analysis comparing difference by groups also after adjusting for age, bmi, sex etc.

Author Response

Response to reviewer #1

We would like to thank the reviewer for his/her constructive criticism and helpful suggestions, which have helped us to revise and improve our manuscript, as well as his/her kind remarks.

 #1. The topic of the paper is relevant, but final results should be evaluated by using multivariate analysis comparing difference by groups also after adjusting for age, bmi, sex etc.

 Thank you very much for identifying this important issue. In response, we considered how to confirm the association between the intake of a specific food factor and the frailty status in the present study. Instead of a multivariate regression analysis, we performed an ANCOVA using a specific food factor as the dependent variable, frailty status (robust, prefrail, or frail) as the independent variable, and age, BMI, smoking habit, and alcohol consumption as independent covariates to exclude the possible confounding effects of baseline differences between the robust, prefrail, and frail groups. This analysis revealed that only vitamin C intake was associated significantly with the frailly status in women diagnosed using the J-CHS criteria (p=0.027), suggesting that age, BMI, a smoking habit, and alcohol consumption were confounding factors affecting the results of ANOVA of female subjects. By contrast, we were able to confirm significant associations of the intakes of soluble dietary fiber, potassium, folate, and vitamin C with the frailty status in men diagnosed using both the J-CHS (p=0.035, 0.023. 0.012, and 0.007, respectively) and KCL criteria (p=0.008, 0.011. 0.007, and 0.006, respectively). Therefore, we concluded that the sufficient intakes of dietary fiber, potassium, folate, and vitamin C may be critical for preventing frailty in men. We have added these analyses and results to the Materials and Methods, Results, and Discussion section of the revised manuscript. However, the abovementioned final result was deleted from the revised abstract due to the word count limitation.

 Thank you again for your excellent suggestions.

Reviewer 2 Report

The manuscript by Tamaki et al. - which aims to evaluate the association between dietary habits and 20 frailty in rural Japanese community-dwelling older adults - is of interest. However, I strongly suggest an extensive editing of English style and language. 

The abstract needs a brief background. The analysis were ultimately conducted on 800 subjects and I suggest to include p-value for significant results.

Although the introduction provides sufficient background and references, I strongly recommend and extensive English editing. 

Methods must be improved. The authors recognized the cross-sectional design as a weakness of their study. Moreover, I strongly suggest to include more details on dietary assessment and questionnaire structure. Statistical analysis must be improved: did the Authors test for normality? The authors have to choose parametric or non-parametric tests according to normality. Moreover, I suggest to include regression analysis, since there are several differences across frailty groups (e.g. age, BMI etc...). 

Presentation of results can be improved. I suggest moderate English revision. Moreover, I recommend to include columns for p-value in all the tables instead of symbols, which make confusion. 

In general, conclusions are supported by the results. However, difference between sexes and diagnostic tools for frailty could be extensively discussed and motivated. 

Author Response

Response to reviewer #2

We would like to thank the reviewer for his/her constructive criticism and helpful suggestions, which have helped us to revise and improve our manuscript. We would also like to thank the reviewer for his/her kind remarks.
#1. The manuscript by Tamaki et al. - which aims to evaluate the association between dietary habits and frailty in rural Japanese community-dwelling older adults - is of interest. However, I strongly suggest an extensive editing of English style and language.

According to the reviewer’s suggestion, we have submitted the revised manuscript for English language editing.

#2. The abstract needs a brief background. The analysis was ultimately conducted on 800 subjects and I suggest to include p-value for significant results.

We have added a brief background, “To develop effective nutritional interventions for preventing frailty, the specific problems associated with the dietary habits of individuals based on sex differences must be identified.” and revised the number of subjects to 800 as indicated. We have also included the p-value in the results and omitted the last sentence, “Sufficient intake of dietary fiber, minerals and vitamins may be critical for the prevention of frailty.” from the revised abstract due to the word count limitation.

#3. Although the introduction provides sufficient background and references, I strongly recommend and extensive English editing.

According to the reviewer’s suggestion, we have submitted the revised manuscript for English language editing.

#4. Methods must be improved. The authors recognized the cross-sectional design as a weakness of their study. Moreover, I strongly suggest to include more details on dietary assessment and questionnaire structure. Statistical analysis must be improved: did the Authors test for normality? The authors have to choose parametric or non-parametric tests according to normality. Moreover, I suggest to include regression analysis, since there are several differences across frailty groups (e.g. age, BMI etc...).

We had described the limitations of our study, including the cross-sectional design of this study, in the first version of our manuscript.

Additionally, we included more details about the BDHQ in the Materials and Methods section of the revised manuscript, as described below.

“Dietary habits during the preceding month were assessed using a brief-type self-administered diet history questionnaire (BDHQ)]. The BDHQ for older people is a 10-page structured questionnaire used to inquire about the consumption frequencies of selected foods commonly consumed in Japan, general dietary behavior and usual cooking methods. The daily intakes of 58 food items, energy, and selected nutrients were calculated BDHQ responses the using an ad hoc computer algorithm based on the Standard Tables of Food Composition in Japan. For all study participants, BDHQ calculations were made using fixed, sex-specific portion sizes. The methods used to calculate the dietary intakes and the validity of the BDHQ have been detailed elsewhere.“

We tested the normality of each result and selected a parametric test.

Thank you very much for identifying this important issue. In response, we considered how to confirm the association between the intake of a specific food factor and the frailty status in the present study. Instead of a multivariate regression analysis, we performed an ANCOVA using a specific food factor as the dependent variable, frailty status (robust, prefrail, or frail) as the independent variable, and age, BMI, smoking habit, and alcohol consumption as independent covariates to exclude the possible confounding effects of baseline differences between the robust, prefrail, and frail groups. This analysis revealed that only vitamin C intake was associated significantly with the frailly status in women diagnosed using the J-CHS criteria (p=0.027), suggesting that age, BMI, a smoking habit, and alcohol consumption were confounding factors affecting the results of ANOVA of female subjects. By contrast, we were able to confirm significant associations of the intakes of soluble dietary fiber, potassium, folate, and vitamin C with the frailty status in men diagnosed using both the J-CHS (p=0.035, 0.023. 0.012, and 0.007, respectively) and KCL criteria (p=0.008, 0.011. 0.007, and 0.006, respectively). Therefore, we concluded that the sufficient intakes of dietary fiber, potassium, folate, and vitamin C may be critical for preventing frailty in men. We have added these analyses and results to the Materials and Methods, Results, and Discussion section of the revised manuscript.

#5. Presentation of results can be improved. I suggest moderate English revision. Moreover, I recommend to include columns for p-value in all the tables instead of symbols, which make confusion.

According to the reviewer’s suggestion, we have submitted the revised manuscript for English language editing.

We also inserted columns listing the p-values at the right sides of all tables in the revised manuscript.

#6. In general, conclusions are supported by the results. However, difference between sexes and diagnostic tools for frailty could be extensively discussed and motivated. 

Thank you very much for your constructive advice. Accordingly, we have described the differences between the tools used to diagnose frailty in the second paragraph of the Discussion in the revised discussion.

“In this study, we used the J-CHS as the phenotype model, and the KCL as the accumulation of deficits model. The KCL was originally designed to screen Japanese community-dwelling elderly adults whose status could potentially progress to the point of requiring long-term care. Satake et al. evaluated the adequacy of a diagnosis of frailty severity based on the KCL criteria or the CHS criteria and reported that the respective sensitivity and specificity of these methods were 70.3% and 78.3% for prefrail individuals and 89.5% and 80.7% for frail individuals. In this study, the two frailty diagnoses had a concordance rate of approximately 60%. As the KCL assesses the seven subdomains, including IADL, physical function, nutrition, oral function, socialization, memory, and mood, the impact of the physical aspect of frailty may become diluted by the domains of cognitive impairment, depression, and social frailty in individuals identified as frail using this model. We speculate that recent nutritional issues may more strongly affect the physical aspect of frailty than the neuropsychiatric and social aspects. Accordingly, the differences in basic characteristics and dietary habits between the robust and prefrail or frail groups, as identified using the two distinct sets of criteria, seemed acceptable. Our results suggest that nutritional interventions should be customized according to the characteristics of the diagnostic tool. Furthermore, uniform nutritional instructions such as increases in the calorie and protein intakes might be redundant for the treatment and prevention of frailty and sarcopenia in older adults.

Although the KCL includes two queries about the nutritional status and three about oral function, the total score is used to diagnose frailty. Therefore, the J-CHS might more sensitively detect abnormal dietary habits in an earlier stage of the frailty cycle. Nutritional items should be extracted them from the KCL prior to reevaluation, given the risk that this comprehensive frailty rating system might underestimate issues with a patient’s dietary habits. Accordingly, in a recent KCL-based study, the authors evaluated the associations between the frequency of protein-rich food intake and the frailty subdomains of KCL. Further studies are needed to conclude which frailty assessment tool can better detect potential nutritional disturbances in community-dwelling older adults.”

According to the reviewer’s suggestion, we also revised the third paragraph of the Discussion in the revised manuscript.

In our study, the nutritional issue had a more remarkable effect in frail men, compared with frail women. Notably, the male subjects included higher numbers of current smokers and alcohol drinkers, compared with the female subjects. The excess intake of these luxury grocery items might affect the nutritional statuses of men, consistent with previous studies demonstrating that alcohol consumption can limit nutrient absorption and bioavailability in the blood. This finding suggests a need for increased daily living guidance concerning these the luxury grocery items, especially for men. By contrast, women exhibited a significantly higher prevalence of sarcopenia and osteoporosis. In other words, a sex-related discrepancy exists between the possible involvement of malnutrition in the development of frailty and the higher prevalence of some downstream hallmarks of frailty.

The population in this study appeared to be healthy and well-nourished and had a relatively high intake of protein. In such a population, the effect of an insufficient nutritional intake on the development of frailty may have been affected by other factors. Additionally, inflammation, oxidative stress, and neurohumoral factors may contribute to the development of frailty. Our results led us to speculate that men were more vulnerable to nutritional issues, whereas comorbidities might have a relatively greater effect on the development of frailty in women. Given the nature of frailty as a complex clinical syndrome comprising low physical activity, exhaustion, body weight loss, and other factors, the nutritional issue identified in this study might be more responsible for these components of frailty, rather than a low skeletal muscle mass or low bone density. We note the metabolisms of both bone and muscle are closely associated with sex hormones. Furthermore, the prevalence of osteoporosis and sarcopenia may be affected by comorbid orthopedic diseases (e.g., osteoarthritis of the knee and lumbar spondylosis), most of which develop independently of the nutritional status. These factors may also be responsible for the sex-related discrepancy between the involvement of malnutrition in the development of frailty and the prevalence of some downstream hallmarks of frailty. Therefore, discussions of sex differences in the pathophysiology of frailty should distinguish the osteoarticular and muscular components of frailty from other components.”

Thank you again for your excellent suggestions.
 Reviewer 3 Report

See attached document if hard to read comments below:

This manuscript explored the relationship between dietary habits and frailty in older Japanese adults living in a rural setting. Dietary habits were assessed using a self-administered questionnaire. The authors used two scales to determine frailty: the Kihon checklist (KCL) and the Japanese version of the Cardiovascular Health Study (J-CHS). The manuscript was well written and was easy to follow. The experimental design, and methods are scientifically sound, and results interpreted correctly. I only have two main concerns and a few minor editing/revision suggestions as outlined below.

Major comments:

The authors reported that there was:

a.     No difference in the prevalence of frailty between men and women.

b.     Prevalence of sarcopenia and osteoporosis was higher in women.

c.    Carbohydrate, nutrients, fiber, total protein and fat intake were lower in frail males only. Statistically significant decrease was measured using the KCL criteria but only similar trends noted using the J-CHS criteria.

Frailty predicts both sarcopenia and osteoporosis. While the prevalence of frailty was the same between men and women in this study, women had higher rates of sarcopenia and osteoporosis. The authors presented the hypothesis that nutritional intake is important for the determination of frailty. Interestingly, poor nutrient intake was only found in men. If dietary habits are important to the development of frailty, then the prevalence of frailty or its downstream hallmarks (sarcopenia and osteoporosis) should have been higher in males. The authors should discuss what the implications of this are as the results are in direct contrast to their central hypothesis.

2. Page 15 lines 278-284. The authors suggest nutritional interventions should be targeted to frail men as they suffered from malnutrition compared to frail females.  The authors suggested the underlying reason is likely that men do not cook, or pay attention to meal contents. I disagree with this. If women are cooking the meals, are they making a separate meal for the men vs. themselves? If they are eating the same food, then how are the men suffering from malnutrition and women are not? Perhaps it is less to do with nutrient intake and more to do with the higher prevalence of alcoholism in men. Studies have shown that alcohol consumption can limit nutrient absorption and bioavailability in blood (see https://pubs.niaaa.nih.gov/publications/aa22.htm).

Minor comments:

Page 5, line 151: It was hard to figure out what the numbers represented in the tables initially. The legend for table 1 (and indeed for all tables) can be more explicit and state clearly that “the data in the table represent the prevalence of a condition by the total number of individuals, with % of population in parenthesis next to it. This applies to all variables unless specific units are provided”.

Sometimes the row title in the table doesn’t match the data in the table. This happens in the following instances:

a.     Page 4, table 1, row with frailty diagnosis by KCL criteria: robust, prefrail and frail don't line up with the data in the columns next to it.

b.     Page 5, table 2, smoking habit row needs to be fixed.

c.      Page 8, table 3, smoking habit row needs to be reformatted.

d.     Page 11, table 5, smoking habit row needs to be reformatted.

3.    Page 5, table 2: Consider having the column headers repeated on page 6 as the table is too long to fit on one page. Same with table 3 on page 8/9,  and table 4 on page 11/12.

4.     Page 15, line 272: spelling mistake (redundant).

5.     Page 15, lines 203-205, 208-209: font type/size changed for certain words in these lines (e.g. dietary fiber, longevity, significance of consuming...).

6.     Page 16-17: font type and size for references are different than manuscript.

Author Response

Response to reviewer #3

We would like to thank the reviewer for his/her kind remarks, as well as constructive criticism and helpful suggestions, that have helped us to revise and improve our manuscript.
Major comments:

The authors reported that there was:

No difference in the prevalence of frailty between men and women.

Prevalence of sarcopenia and osteoporosis was higher in women.

Carbohydrate, nutrients, fiber, total protein and fat intake were lower in frail males only. Statistically significant decrease was measured using the KCL criteria but only similar trends noted using the J-CHS criteria.

Frailty predicts both sarcopenia and osteoporosis. While the prevalence of frailty was the same between men and women in this study, women had higher rates of sarcopenia and osteoporosis. The authors presented the hypothesis that nutritional intake is important for the determination of frailty. Interestingly, poor nutrient intake was only found in men. If dietary habits are important to the development of frailty, then the prevalence of frailty or its downstream hallmarks (sarcopenia and osteoporosis) should have been higher in males. The authors should discuss what the implications of this are as the results are in direct contrast to their central hypothesis.

Thank you very much for identifying an important issue. As noted, we hypothesized that the nutritional intake is important to the determination of frailty. However, we identified a close correlation of the nutritional issue with frailty only in men whereas a significant higher prevalence of sarcopenia and osteoporosis was observed in women. In other words, we have identified sex-related discrepancies in the possible involvement of malnutrition on frailty and the prevalence of some downstream hallmarks of frailty. This discrepancy should be discussed from the perspective of complexity. First, the population in this study seemed healthy and well-nourished and therefore the effect of a nutritional issue on the development of frailty may have been obscured by other factors, such as oxidative stress, and neurohumoral factors. Our results suggest that men are more vulnerable to a nutritional issue and this phenomenon may be partly attributable to a higher rate of alcohol consumption and higher prevalence of smoking. By contrast, we concluded that the influence of comorbidities might have a larger impact on the development of frailty in women than in men. Second, sex hormones are closely associated with the metabolism of both bone and muscle, and several previous epidemiological studies have demonstrated sex-related differences in the prevalence of osteoporosis and sarcopenia. Finally, frailty is a complex clinical syndrome that includes low physical activity, exhaustion, and body weight loss. The nutritional issue identified in our study might be responsible for these components of frailty, rather than a low skeletal muscle mass or low bone density. Therefore, discussions of sex differences in the pathophysiology of frailty should distinguish the osteoarticular and muscular components of frailty from other components. We have added a discussion of this aspect to the revised text.

Page 15 lines 278-284. The authors suggest nutritional interventions should be targeted to frail men as they suffered from malnutrition compared to frail females.  The authors suggested the underlying reason is likely that men do not cook, or pay attention to meal contents. I disagree with this. If women are cooking the meals, are they making a separate meal for the men vs. themselves? If they are eating the same food, then how are the men suffering from malnutrition and women are not? Perhaps it is less to do with nutrient intake and more to do with the higher prevalence of alcoholism in men. Studies have shown that alcohol consumption can limit nutrient absorption and bioavailability in blood (see https://pubs.niaaa.nih.gov/publications/aa22.htm).

Thank you very much for identifying this important issue. For older adults living alone, our hypothesis that men do not cook or pay attention to meal contents may explain, at least in part, the malnutrition observed in men. However, we did not ask the participants about their living status, as mentioned in the description of limitations. Although we did not know the exact number of older participants who lived alone, the majority seemed to live in couples or with family. Therefore, we have corrected the discussion regarding sex differences according to the reviewer’s suggestion, as shown below.

“In our study, the nutritional issue had a more remarkable effect in frail men, compared with frail women. Notably, the male subjects included higher numbers of current smokers and alcohol drinkers, compared with the female subjects. The excess intake of these luxury grocery items might affect the nutritional statuses of men, consistent with previous studies demonstrating that alcohol consumption can limit nutrient absorption and bioavailability in the blood. This finding suggests a need for increased daily living guidance concerning these the luxury grocery items, especially for men.”

Minor comments:

Page 5, line 151: It was hard to figure out what      the numbers represented in the tables initially. The legend for table 1      (and indeed for all tables) can be more explicit and state clearly that “the data in the table represent the prevalence of a      condition by the total number of individuals, with % of population in      parenthesis next to it. This applies to all variables unless specific      units are provided”.

Thank you very much for your constructive advice. We have accordingly included this legend in the revised manuscript.

“The data in the table represent the prevalence of a condition by the total number of individuals (Mean±SD or number), with % of population in parenthesis next to it. This applies to all variables unless specific units are provided.”

Sometimes the row title in the table doesn’t      match the data in the table. This happens in the following instances:

Page 4, table 1, row with frailty diagnosis by       KCL criteria: robust, prefrail and frail don't line up with the data in       the columns next to it.

Page 5, table 2, smoking habit row needs to be       fixed.

Page 8, table 3, smoking habit row needs to be       reformatted.

Page 11, table 5, smoking habit row needs to be       reformatted.

We corrected these issues in the revised manuscript.

Page 5, table 2: Consider having the column      headers repeated on page 6 as the table is too long to fit on one page.      Same with table 3 on page 8/9, and table 4 on page 11/12.

We corrected these issues in the revised manuscript.

Page 15, line 272: spelling mistake (redundant).

We corrected this spelling error in the revised manuscript.

Page 15, lines 203-205, 208-209: font type/size      changed for certain words in these lines (e.g. dietary fiber, longevity,      significance of consuming...).

We corrected these inconsistencies in the font type/size in the revised manuscript.

Page 16-17: font type and size for references are      different than manuscript.

We corrected these inconsistencies in the font type/size in the revised manuscript.
Thank you again for your excellent suggestions.

Round  2

Reviewer 1 Report

Dear Authors

the paper is substantially improved in this form

Reviewer 2 Report

The Authors accomplished all my previous comments and suggestions